# Selective Axillary Dissection after Neoadjuvant Chemotherapy in Patients with Lymph-Node-Positive Breast Cancer (CLYP Study): The Radio-Guided Occult Lesion Localization Technique for Biopsy-Proven Metastatic Lymph Nodes

**DOI:** 10.3390/cancers15072046

**Published:** 2023-03-29

**Authors:** Rossella Rella, Marco Conti, Enida Bufi, Charlotte Marguerite Lucille Trombadori, Alba Di Leone, Daniela Terribile, Riccardo Masetti, Luca Zagaria, Antonino Mulè, Francesca Morciano, Gianluca Franceschini, Paolo Belli

**Affiliations:** 1UOC di Radiologia Toracica e Cardiovascolare, Dipartimento di Diagnostica per Immagini, Radioterapia Oncologica ed Ematologia, Fondazione Policlinico Universitario Agostino Gemelli IRCCS, Largo A. Gemelli 8, 00168 Rome, Italy; 2Centro Integrato di Senologia, Dipartimento di Scienze della Salute della Donna e del Bambino e di Sanità Pubblica, Fondazione Policlinico Universitario Agostino Gemelli IRCCS, Largo A. Gemelli 8, 00168 Rome, Italy; 3UOC di Medicina Nucleare, Dipartimento di Diagnostica per Immagini, Radioterapia Oncologica ed Ematologia, Fondazione Policlinico Universitario Agostino Gemelli IRCCS, Largo A. Gemelli 8, 00168 Rome, Italy; 4Dipartimento Scienze della Salute della Donna e del Bambino e di Sanità Pubblica, Unità di Gineco-Patologia e Patologia Mammaria, Fondazione Policlinico Universitario Agostino Gemelli IRCCS, Largo A. Gemelli 8, 00168 Rome, Italy; 5Facoltà di Medicina e Chirurgia, Università Cattolica Sacro Cuore, Largo F. Vito 1, 00168 Rome, Italy

**Keywords:** clipped lymph node, neoadjuvant systemic therapy, node-positive breast cancer patients, Radio-Guided Occult Lesion Localization (ROLL) technique, sentinel lymph node biopsy

## Abstract

**Simple Summary:**

The aim of the study was to evaluate the accuracy of the Radio-Guided Occult Lesion Localization (ROLL) technique for biopsy-proven metastatic axillary lymph nodes in nodal staging after neoadjuvant chemotherapy in patients with node-positive breast cancer at diagnosis. The ROLL procedure for metastatic axillary lymph nodes, identified with a clip marker placement before neoadjuvant chemotherapy initiation, demonstrated an improvement in detection of residual axillary disease in comparison with sentinel lymph node biopsy alone.

**Abstract:**

(1) Background: To help to refine the accuracy of sentinel lymph node biopsy (SLNB) in breast cancer (BC) patients with biopsy-proven nodal disease prior to neoadjuvant chemotherapy (NACT), a method of marking the biopsy-proven positive LN at diagnosis to enable its removal during surgery was proposed. The aim of this study was to evaluate the accuracy of the Radio-Guided Occult Lesion Localization (ROLL) technique of biopsy-proven metastatic LN in nodal staging after NACT among node-positive BC patients. (2) Methods: Patients with invasive BC and biopsy-proven axillary metastases receiving NACT were enrolled. A clip marker was placed on the sampled LN (clipped lymph node, CLN) before NACT. Before surgery, the ROLL procedure (radioactive tracer injection into CLN under ultrasound guidance) was performed, and the CLN was surgically resected. The correspondence between the CLNs and SLNs was evaluated. The pathologic findings of the CLNs and SLN(s) were compared with remaining axillary nodes at ALND to determine false negative rates (FNRs). (3) Results: Seventy-two patients were analyzed. Surgery successfully identified the CLN in 70/72 procedures (97.2%). For 60/72 patients who underwent ALND, the FNRs dropped from 19.35% for SLNB to 3.13% for CLN biopsy. (4) Conclusions: The ROLL procedure got CLNs is accurate in axillary nodal staging after NACT in node-positive BC patients at diagnosis.

## 1. Introduction

Today, the axillary lymph node (LN) status is one of the most important prognostic factors for breast cancer [1]. It is used to determine the cancer stage and guides treatment planning [2]. Management of the axilla in patients with breast cancer has been changing rapidly over the last few years. In patients presenting with biopsy-proven nodal disease prior to neoadjuvant chemotherapy (NACT), the standard surgery for treating the axilla has long been axillary lymph node dissection (ALND) [3]. However, modern chemotherapy regimens can lead to the eradication of LN disease in approximately 40% of patients, with higher percentages reached for some biological cancer subtypes (74% in the case of human epidermal growth factor receptor 2-positive disease and 49% in triple-negative breast cancer) [4]. Given the high disease eradication rate of breast tissues and axillary LNs and the great success of the sentinel lymph node biopsy technique (SLNB) for node-negative cancers treated with NACT, SLNB was also proposed as a feasible surgical strategy for breast cancer patients with nodal involvement at diagnosis in order to prevent complications of ALND [5]. However, in these patients, SLNB proved to be insufficiently accurate in the restaging of the axilla and identifying patients with complete LN disease regression after NACT, with unacceptable false negative rates (FNRs) ranging from 12.6% to 24.3% [6,7]. In particular, in the four prospective trials (ACOSOG Z10716, SENTINA [7], SN FNAC [8], and the GANEA 2 trial [9]), the FNRs failed to meet the threshold of 10% that is considered clinically satisfactory, with FNRs ranging from 11.9% [9] to 14.2% [7].

Several studies have proposed different ways to refine SLNB’s accuracy among these patients and achieve a more reliable definition of complete nodal pathologic response. These include the mandatory use of immunohistochemistry [8], the resection of more than one sentinel lymph node (SLN) (when feasible) [10], the use of the dual mapping technique (both blue dye and radiolabeled colloid mapping agents) [6,7], and the marking of the biopsy-proven positive lymph nodes at diagnosis to enable its removal during surgery [11,12,13,14].

In particular, concerning the latter strategy, in an additional subset analysis of the ACOSOG1071 trial, the FNR was reduced to 6.8% (95% CI: 1.9%, 16.5%) when the metastatic LN with the clip was retrieved and excised during surgery [11]. A marker is therefore placed on the biopsy-proven positive LN before NACT initiation so as to enable its removal during surgery, using the marker as a target for preoperative localization. Another study on the selective evaluation of clipped nodes showed that the FNR was reduced to 4% and further lowered to approximately 2% when associated with SLNB, a surgical technique called targeted axillary dissection [12]. This strategy showed promising results, enabling practitioners to work around the problems of altered lymphatic drainage with chemotherapy and the non-sequential response of the involved ALNs to NACT, which could lead to FN results of SLNB and unsuccessful lymphatic mapping [13].

Therefore, different techniques have been proposed to localize and enable the surgical excision of the marked LN, such as hook wire localization [14], intraoperative ultrasound-guided excision [15], tattooing with a sterile black carbon suspension [16,17], the use of magnetic seeds [18] or radar markers [19], and radioactive iodine I 125–labeled titanium seed localization [20,21].

The aim of this study was to evaluate the accuracy of the Radio-Guided Occult Lesion Localization (ROLL) technique for biopsy-proven metastatic LNs in nodal staging after NACT in patients with node-positive breast cancer at diagnosis.

## 2. Materials and Methods

This single-institution prospective study was approved by our institutional review board and by our ethics committee. All patients provided written informed consent.

### 2.1. Study Population

Patients with (a) invasive breast cancer confirmed by preoperative image-guided core-needle biopsy (CNB) and (b) axillary metastases confirmed by CNB who (c) were receiving NACT were offered to participate in the trial. Patients with distant metastases at diagnosis, previous breast cancer, previous breast or axillary surgery, pregnancy, or breast cancer of stage T4 were excluded.

A total of 104 patients were initially enrolled from February 2018 to February 2020. Clinical–pathologic data were collected from the patients’ medical records and recorded.

When nodal metastasis was demonstrated by CNB of a suspicious LN, a clip marker—a HydroMARK^®^ clip (Mammotome, Cincinnati, OH, USA) or CorMARK (Devicor, Cincinnati, OH, USA)—was placed within the cortex of the sampled LN (clipped lymph node, CLN) before the beginning of NACT.

Before surgery (from 24 to 3 h before surgery), an experienced breast radiologist performed an US axillary evaluation to determine the axillary status after NACT and identify the CLN. Then, the ROLL of the CLN was performed. In the ROLL technique, a radioactive tracer—specifically, Technetium 99 mTc macro-aggregated albumin (99mTc-MAA; activity min-max: 18–37 MBq; Volume: 0.2–0.4 mL) —is injected into the CLN under US guidance using a 21G needle. At the time of surgery, the ROLL-localized CLN was resected first, followed by the removal of the remaining SLNs. A hand-held gamma probe was used to guide the intra-operative identification of the ROLL-marked CLN and its surgical resection. The presence of the pre-NAC-placed biopsy clip within the localized LN was immediately confirmed upon specimen radiograph. SLNB was performed using the blue dye method [22]. The CLN, SLN(s), and any additional resected lymph nodes were identified and individually sent for pathological examination. ALND was always performed after the pathological examination of the SLN/CLN showed a tumor-positive result, while for the patients with a complete clinical response and tumor-negative SLN/CLNs, the surgeon could choose to omit ALND and perform only nodal sampling.

### 2.2. Pathological Evaluation

The SLNs, CLN, and all other resected LNs were submitted fresh for intraoperative macroscopic and cytologic evaluation. The intraoperative cytology examination of the lymph nodes was performed by dissecting them into two parts along the long axis. After sectioning, a slide was applied or dragged over the cut surface of both halves, and the sample was stained with Harris hematoxylin solution. In cases of suspected cytology, the lymph node halves were frozen to −22 °C, serially divided into ultrathin sections, and stained with Harris hematoxylin solution. In cases of negative findings, the LNs were used for complete sectioning in steps of 200 mm, with 2 consecutive slides stained with hematoxylin and eosin, and pan-cytokeratin (AE1/AE3) immunohistochemistry. The LN metastasis classification followed the recommendations of the 7th American Joint Committee on Cancer (2010) [23], after NACT was used to isolate tumor cells and the micrometastases were considered node-positive. The histopathologic results of CLN, SLNs, and all the resected LNs were given separately in the final pathology report.

### 2.3. Statistical Analysis

The identification rates of CLN and SLN(s) were evaluated. The number of patients for whom a CLN was found in the SLNs was summarized with counts and relative frequencies.

In patients who underwent ALND, the pathologic findings regarding the CLN and in SLN(s) were compared with the remaining resected axillary nodes to determine the FNR. The FNR was calculated as the number of cases in which the specified node (the CLN or the SLN) did not show metastasis, even though residual disease was seen in other axillary nodes, divided by the total number of pathologically node-positive patients. Receiver Operating Characteristic (ROC) curves were constructed to visualize the diagnostic effectiveness of the described method (CLN and ROLL) in order to correctly identify the SLN based on its final histology. The area under the ROC curve (AUC) was computed, and the curves were presented in the binormal form.

## 3. Results

Patients who did not undergo surgical treatment after NACT (the discovery of M1-disease during NACT) or who underwent surgery at another institution (14 out of 104 patients, 13.5%), those who discontinued NACT (3 out of 104 patients, 2.9%), and those who did not undergo the ROLL procedure used to identify clip-marked lymph nodes due to patient refusal (6 out of 104 patients, 5.8%) or did not complete the intended protocol due to other reasons (4 out of 104 patients, 3.8%) were excluded. In 5 out of 77 patients (6.5%), the clip was not sonographically identified, and the ROLL procedure was not performed. For all these patients, ALND was performed, with residual nodal disease observed in two patients. In the remaining 72 procedures (72/77, 93.5%), the ROLL node injection was performed under US guidance (Figure 1).

Finally, 72 patients who actually underwent the ROLL procedure were included in the study, with a median age at the time of enrolment of 51.4 (range 28–76). The characteristics of the study population are shown in Table 1.

Figure 2 shows the flow chart of the study.

An SLN was identified with the blue dye method in 66 out of 72 patients (91.7%), and the mean number of SLNs retrieved was 2.6 (range 1–7). Surgery successfully identified the ROLL-marked CLN in 70 of 72 procedures (97.2%) (Figure 3).

In the two patients with unsuccessful identification of the CLN, the ROLL procedure identified a tumor-negative LN without a clip, in one case corresponding to a colored SLN. Both sets of patients underwent ALND, and all the resected LNs underwent specimen radiography, with the CLN retrieved from among the other resected nodes. The CLN was documented to be one of the SLNs in 51 of 66 patients with an SLN identified (77.3%). When using the two methods combined, the detection rate increased to 98.6%. Table 2 shows the CLN and SLN detection rates according to each modality technique (blue dye and ROLL).

### 3.1. CLN and SLNs and the Prediction of the Nodal Status

The FNR was evaluated in 60 out of 72 patients who underwent ALND. The nodal pathologic complete response (pCR) rate at ALND was 46.67% (28 out of 60 patients without residual disease). The remaining 12 patients underwent nodal sampling, with a mean number of resected nodes of 4. In these patients, only tumor-negative LNs were found, with signs of response upon histopathological assessment; therefore, ALND was omitted.

No SLN was identified in five patients who had ALND, of whom one had residual nodal disease. Among the remaining 31 patients with residual disease who had an SLN identified, in 6 of these patients, the SLN did not show metastasis, but residual metastatic disease was demonstrated elsewhere upon ALND, resulting in an FNR of SLND of 19.35% (6 false negative events in 31 patients with residual nodal disease).

CLN was not correctly identified through the ROLL procedure in two patients who had ALND. The CLN, retrieved from among the other resected nodes, showed a negative result in both cases, without residual nodal disease in the other nodes. Among the 32 patients with residual nodal disease, in one case, the CLN did not show metastasis, but residual metastatic disease was demonstrated upon ALND, resulting in an FNR of CLN of 3.13% (1 false negative event among 32 patients with residual nodal disease). In particular, in five of the six patients with FN SLNs, the CLN contained metastatic disease. The false negative rates of CLN, SLN, and both methods combined are shown in Table 3.

The ROC curve for the detection of SLN is displayed in Figure 4.

The corresponding AUC was 0.787, indicating an adequate diagnostic performance (*p* < 0.001, 95% confidence limit 0.65 to 0.87). The diagnostic accuracy was 78%.

### 3.2. Factors Contributing to Prediction of Nodal Status for CLN and SLNs

In 60 patients who underwent ALND, the prediction of nodal status of the CLN and SLN(s), according to the number of abnormal LNs upon US axillary evaluation at diagnosis and US evaluation after NACT, was assessed (Table 4).

Basing on the nodal burden (<4 abnormal LNs versus ≥4 abnormal LNs) upon axillary US evaluation before NACT, 35 patients in the first group (35/43, 81.4%) and 25 in the second group (25/29, 86.2%) underwent ALND. The FNRs of SLND were 31.25% (5 negative SLNs out of 16 patients with residual disease) and 6.67% (1 false negative event among 15 patients with residual disease) in the two groups, respectively. The CLN did not show metastasis in one case with residual disease (FNR 5.88%) in the first group upon ALND, while no false negative events were reported in the second group.

In patients with a normal LN status upon post-NACT axillary US, the SLNs did not show metastasis in three patients with residual nodal disease upon ALND, with an FNR of SLND of 30.0% (three false negative events among 11 patients with residual disease and one case in which SLN was not identified), while in patients with suspicious LNs upon post-NACT US, the FNR of SLND was 14.29% (three false negative events among 21 patients with residual disease). In two of the three patients in the first group and in all three patients in the second group with FN SLNs, the CLN showed metastatic disease, with FNRs of CLN of 9.09% and 0%, respectively.

## 4. Discussion

Our results confirm that the ROLL-based localization of CLNs is a feasible approach, and its evaluation in patients with breast cancer and axillary LN involvement at diagnosis improves the detection of residual axillary disease, resulting in an FNR of 3.13%.

This technique of localization successfully identified the CLN in 97.2% of patients, in line with a recently published work [24]. However, the previously cited study did not use specimen radiography to immediately confirm the clip’s removal, which allows the surgeon to gain prompt feedback on the procedure’s outcome. Nevertheless, in our experience, the major limitation of this technique is the US identification of the clip marker, In particular, when the normalization of the LN architecture occurs after NACT, the hyperechogenicity of the clip may not be visible, especially when the clip is located in the LN hilum. Indeed, even if HydroMARK^®^ clips have been demonstrated to retain their visibility after 12 weeks of follow-up [25], their linear shape, associated with the relatively long duration of NACT, together with embedding material reabsorption, can make detection of the marker challenging. To work around this issue, it can be useful to employ different types of clips, such as the UltraCor Twirl type that, in the authors’ experience, can be easily visualized due to its particular shape [26]. The other drawback of this technique is the double-step approach (clip positioning before NACT and clip localization after NACT), and it would be desirable to use a one-step procedure. In this regard, iodine-125 seeds (MARI procedure) [20,21] are a possibility or, even better, the use of magnetic seeds [18] or radar markers [19], which also avoid the use of radioactive tracers. Additionally, carbon tattooing was demonstrated to be a simple and less expensive method, with no necessity for additional radiological imaging or nuclear medicine procedures for localization [16,17,27]. However, ROLL is a feasible and effective method for hospitals that cannot use iodine-125 or magnetic seeds.

The optimal staging and management of the axilla in breast cancer patients with axillary nodal involvement at diagnosis who receive NACT is still controversial. Our results confirm that the removal of biopsy-proven positive CLNs improves the diagnostic accuracy of nodal staging after NACT. In our series, the CLN was not identified as an SLN using the traditional mapping technique in 22.7% of patients, in line with previous results (23% in the MD Anderson Cancer Center prospective study and 20% in the Z1071 trial) [11,12].

As previously reported, this procedure (the removal of biopsy-proven metastatic LNs) decreases the FNRs (3.13% in our series) with respect to SLND alone [11,12]. This can allow for a more complete assessment of the nodal response to NACT, enabling the de-escalation of the surgical procedure of the axilla for node-positive breast cancer patients treated with NACT while avoiding significant potential consequences, such as the misrepresentation of the prognosis and non-resection of LNs showing potentially chemotherapy-resistant disease, with long-term consequences [28]. Different from previous published results of the combination of CLN biopsy and SLNB [12,29,30], our results showed that the CLN technique’s accuracy is not further improved when it is combined with the SLN procedure. Indeed, we do not have cases of positive SLNs and negative CLNs in patients with residual disease upon ALND. This result is probably related to the relatively high FNR of SLNB in our series (18.75%). Evidence has demonstrated that fluorescence imaging for axillary SLN identification with indocyanine green or double labelling are superior to the single technique using blue dye [31,32], and prospective studies [6,7,8] have shown that an FNR of <10% can be achieved if more than two SLNs are excised, a dual-tracer technique is used, and isolated tumor cells are considered as positive nodes. In our study, a single-tracer technique was used and, even if an SLN was identified in 66 out of 72 patients (91.7%), being within the range reported in previous studies [6,7,33], this probably affected the FNR of SLNB. However, the CLN procedure alone is able to retain an FNR under the 10% threshold, which is acceptable for clinical care. The ROC AUC for SLN identification was 0.787, and the accuracy was 78%. While these results are consistent and indicate adequate accuracy, improvements in performance need to be achieved before the clinically evaluating the SLN/ROLL procedure to avoid ALND. Additional predictors should be explored to identify patient subsets for whom this approach could be even more effective from a diagnostic point of view.

We also explored the factors that contributed to the prediction of the nodal status of CLNs (and SLN) upon ALND in order to select patients that could benefit most from this procedure. In particular, axillary US evaluation after the completion of NACT was previously proposed to reduce the unacceptably high FNR of SLNB. After NACT, the normalization of the morphologic US characteristics of LNs was associated with a higher probability of pCR [34], and it was demonstrated that axillary US should be performed preoperatively after NACT to optimally identify patients who can be offered SLNB alone as an alternative to ALND. The Current National Comprehensive Cancer Network guidelines state that in NAC, SLNB alone may be permitted for selected patients if a positive axilla is resolved with therapy, but to date, the rate of regional recurrence in patients with proven nodal metastases who have a pCR and undergo SLNB alone is uncertain [28]. Indeed, the FNR of SLNB among patients who have normal lymph node status upon post-chemotherapy US remains high (15.0% versus 8.1% among patients with suspicious nodal status) [35]. Our results are in line with the previously cited work (FNRs of SLNB in patients with normal vs. suspicious LN status upon post-chemotherapy US of 30.0% and 14.29%, respectively) and, in addition, we found that the CLN procedure is effective also in the first category (normal LN status upon post-chemotherapy US), improving the diagnostic performance of nodal staging (FNR = 9.09%) and helping practitioners to choose patients for whom ALND can be safely omitted. However, it must be highlight that the impact of the FNR of SLNB on oncologic outcomes is still unclear and does not seem to correspond to an increase in axillary recurrences or worsening of disease-free survival or overall survival [36]. Ongoing studies, such as AXANA (a large, prospective, non-interventional cohort study), aim to demonstrate which is the best method of axillary surgery after NACT in cases starting with a positive axilla, with the primary endpoints being invasive-disease-free survival, the axillary recurrence rate, quality of life, and arm morbidity [37].

The limitations of our study include its small sample size, performance at a single comprehensive cancer institution, and the single-tracer technique used for SLN identification. Moreover, ALND, which allows for the determination of the FNR, was performed on 60 out of 72 patients, while the remaining 12 patients only underwent nodal sampling.

## 5. Conclusions

The ROLL procedure for CLN can be proposed for axillary nodal staging after NACT in patients with node-positive breast cancer at diagnosis, with an FNR of 3.13%. The CLN procedure is useful when the positive axilla is resolved with therapy. Additional investigations are required to confirm our finding that the ROLL procedure is beneficial for making further treatment decisions and identifying subsets of patients for whom ALND can be safely omitted based on this approach.

## Figures and Tables

**Figure 1 cancers-15-02046-f001:**
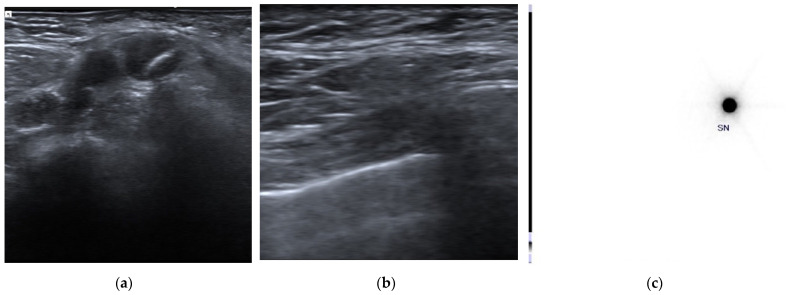
Clip placement and pre-surgical identification of clipped axillary lymph node. (**a**) A clip marker was placed within the cortex of the histologically confirmed metastatic axillary lymph node (clipped lymph node, CLN) before the beginning of neoadjuvant chemotherapy. (**b**) Before surgery, an US axillary evaluation was performed to identify the CLN and a radioactive tracer—Technetium 99mTc macro-aggregated albumin (99mTc-MAA)—was injected into the CLN under US guidance using a 21G needle. (**c**) Pre-surgical scintigraphic acquisition shows the presence of focal activity in the tracer inoculation site.

**Figure 2 cancers-15-02046-f002:**
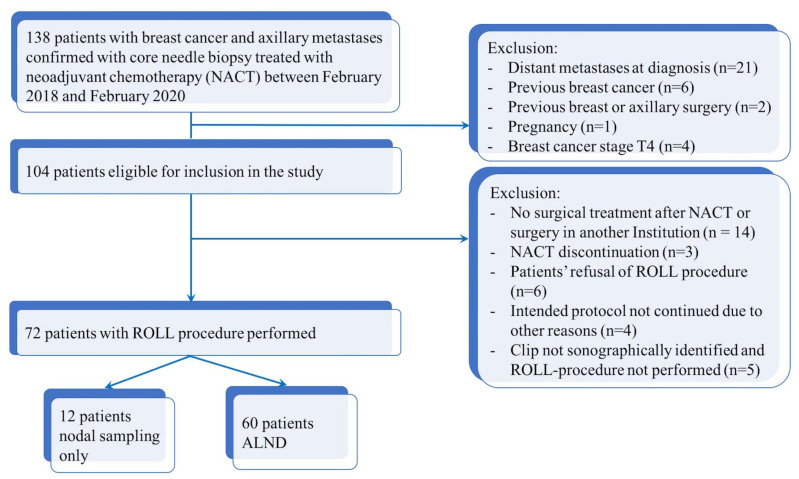
Flow chart diagram of patient selection.

**Figure 3 cancers-15-02046-f003:**
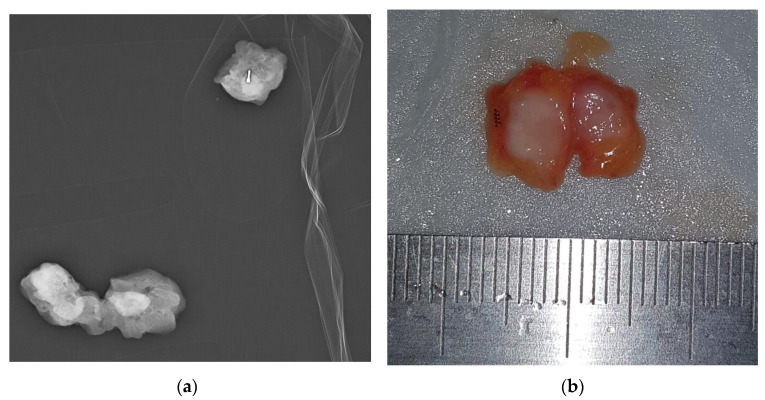
ROLL-localized removal of clipped lymph node. (**a**) Once the localized node is surgically removed, a specimen radiograph is performed to ensure that the clip has been removed. (**b**) The tracer was injected for localization. Then, ROLL of the CLN was performed. In the ROLL technique, a radioactive tracer—0.2 mL of Technetium 99mTc macro-aggregated albumin (99mTc-MAA)—is injected into the CLN under US guidance using a 21G needle. The CLN was bisected along the long axis with the clip as evidence.

**Figure 4 cancers-15-02046-f004:**
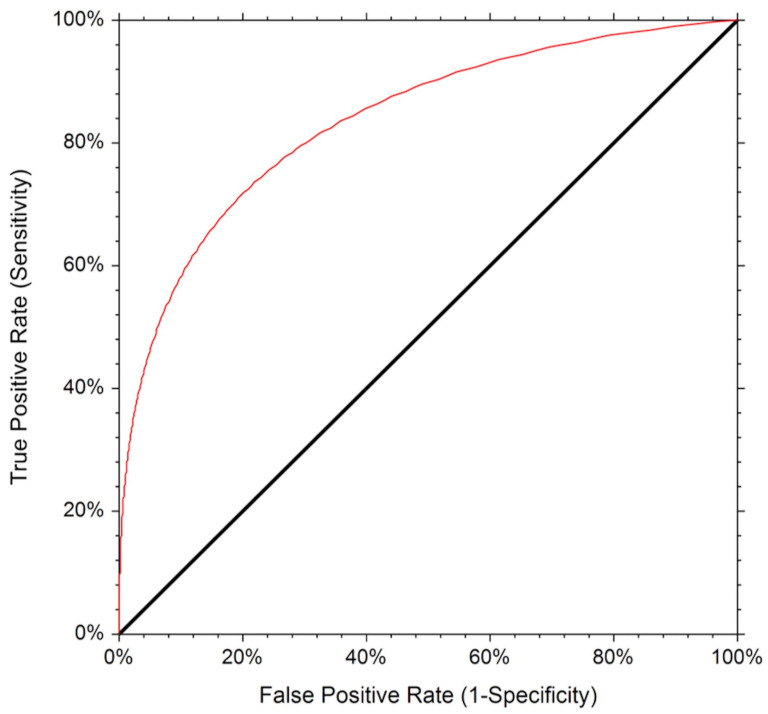
The Receiver Operating Characteristic (ROC) curve for the detection of SLN.

**Table 1 cancers-15-02046-t001:** Patient and tumor characteristics.

**N° of Patients**	**72**
**Mean age, years**	51.4 (28–76)
**Largest tumor diameter on MRI (mm)**	38.3 (12–93)
**Histologic finding**	
Invasive ductal carcinoma	58/72 (80.6%)
Invasive lobular carcinoma	9/72 (12.5%)
Other invasive carcinoma	5/72 (6.9%)
**Stage at diagnosis**	
II	34/72 (47.2%)
III	38/72 (72.8%)
IV	0/72 (0.0%)
**Phenotype**	
HR-negative/HER2-negative	17/72 (23.6%)
HR-positive/HER2-positive	12/72 (16.7%)
HR-negative/HER2-positive	8/72 (11.1%)
HR-positive/HER2-positive	35/72 (48.6%)
**Histologic grade**	
Grade I	3/72 (4.2%)
Grade II	30/72 (41.7%)
Grade III	39/72 (54.1%)
**Type of surgery**	
Breast-conserving surgery	39/72 (54.2%)
Mastectomy	33/72 (45.8%)
**Neoadjuvant chemotherapy regimen**	
Anthracycline plus taxane	26/72 (36.1%)
Anthracycline plus taxane plus anti-HER2	43/72 (59.7%)
Endocrine therapy	3/72 (4.2%)

Numeric data are presented as the median with the range in parentheses. Nonnumeric data are presented as numbers of patients in proportions followed by percentages in parentheses. HER2+, human epidermal growth factor receptor 2-positive; HR, hormone receptor; MRI, Magnetic Resonance Imaging.

**Table 2 cancers-15-02046-t002:** Sentinel lymph node/clipped lymph node detection rate according to each modality technique.

Identification Technique	Identified	Not Identified	Detection Rate (%)
Blue-dye	66	6	91.7
ROLL	70	2	97.2
Blue dye + ROLL	71	1	98.6

ROLL, Radio-Guided Occult Lesion Localization.

**Table 3 cancers-15-02046-t003:** The false negative rates of SLN, CLN, and both methods combined.

	ALND Outcome		
	Residual Disease (n = 32)	No Residual Disease (n = 28)	False Negative Rate (%)
**SLN**			
Positive	25	0	
Negative	6	24	
Not colored	1	4	
			6/31 (19.35%)
**CLN**			
Positive	31	0	
Negative	1	26	
Not identified by ROLL	0	2	
			1/32 (3.13%)
**SNL and CLN**			
SLN-positive/CLN-positive	25	0	
SLN-negative/CLN-positive	5	0	
SLN-positive/CLN-negative	0	0	
SLN-negative/CLN-negative	1	22	
SLN not colored/CLN-positive	1	0	
SLN not colored/CLN negative	0	4	
SLN-negative/CLN not identified	0	1	
SLN not colored/CLN not identified	0	1	
			1/32 (3.13%)

CLN, clipped lymph node; SLN, sentinel lymph node.

**Table 4 cancers-15-02046-t004:** Pathologic results of clipped lymph nodes and sentinel lymph node(s) stratified according to abnormal nodes on initial staging by ultrasound and ultrasound evaluation after neoadjuvant chemotherapy.

**Number of Abnormal Nodes on Initial Staging by US**						
	**<4 Abnormal Nodes**			**≥4 Abnormal Nodes**		
	**ALND Outcome**			**ALND Outcome**		
	**Residual Disease**	**No Residual Disease**	**FNR** **(%)**	**Residual Disease**	**No Residual Disease**	**FNR (%)**
**Outcome CLN**						
Positive	16	0		15	0	
Negative	1	18		0	10	
			1/17 (5.88%)			0/15 (0%)
**Outcome SLNs**						
Positive	11	0		14	0	
Negative	5	17		1	7	
Not identified	1	1		0	3	
			5/16 (31.25%)			1/15 (6.67%)
**Ultrasound Evaluation after NACT**						
	**Normal Lymph Node Status**			**Suspicious Nodal Status**		
	**Outcome ALND**			**Outcome ALND**		
	**Residual Disease**	**No Residual Disease**	**FNR** **(%)**	**Residual Disease**	**No Residual Disease**	**FNR** **(%)**
**Outcome CLN**						
Positive	10	0		21	0	
Negative	1	24		0	4	
			1/11 (9.09%)			0/21 (0%)
**Outcome SLNs**						
Positive	7	0		18	0	
Negative	3	21		3	3	
Not identified	1	3		0	1	
			3/10 (30.0%)			3/21 (14.29%)

ALND, axillary lymph node dissection; CLN, clipped lymph node; NACT, neoadjuvant chemotherapy; SLN, sentinel lymph node; US, ultrasound.

## Data Availability

Data available on request due to privacy restrictions.

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
