# Peer review of "Selective Axillary Dissection after Neoadjuvant Chemotherapy in Patients with Lymph-Node-Positive Breast Cancer (CLYP Study): The Radio-Guided Occult Lesion Localization Technique for Biopsy-Proven Metastatic Lymph Nodes"

_cancers, 2023, doi:10.3390/cancers15072046_

Round 1

Reviewer 1 Report

With the increasing of early-stage breast cancer sentinel lymph node biopsy (SLNB), a minimal invasion method, had comparable accuracy with axillary lymph node dissection (ALND) in lymph node staging. However, SLNB was proved to have a not acceptable false negative rate (FNRs) in patients with neoadjuvant chemotherapy (NACT). Thus, it is extremely meaningful to help refine SLNB accuracy.

Analysis is well conducted and many data are present in this review. Nevertheless, I have some small suggestion:

1.   It seemed that the full name of “FNRs” was not given in Simple Summary when it first proposed. (In Simple Summary)

2.   Maybe, we’d better use immunohistochemistry staining for further identification of resected LNs. (In 2.2 Pathological evaluation)

3.   Evidence demonstrated that fluorescence imaging for axillary SLN identification with indocyanine green (ICG) or double labelling are superior to the single technique with blue-dye (In 2.1 Study population) ([1, 2]). If available, using double tracer technique as control group would be better in further study.

4.   In addition, whether the ROLL is benefit for the further treatment decision needed further study.

 1.       Kedrzycki, M.S., et al., Meta-analysis Comparing Fluorescence Imaging with Radioisotope and Blue Dye-Guided Sentinel Node Identification for Breast Cancer Surgery. Ann Surg Oncol, 2021. 28(7): p. 3738-3748.

 2.       Almhanedi, H., et al., Novel double injection technique for sentinel lymph node biopsy in oral cancer. Br J Oral Maxillofac Surg, 2021. 59(10): p. 1296-1301.

Author Response

Dear Editor,

Thank you for your email enclosing the reviewers’ comments.

We have carefully reviewed the comments and have revised the manuscript accordingly. Our responses are given in a point-by-point manner below.

The authors would like to thank the reviewers for their precious time and invaluable comments.

We really hope these modifications can meet with your approval.

REVIEWER 1

With the increasing of early-stage breast cancer sentinel lymph node biopsy (SLNB), a minimal invasion method, had comparable accuracy with axillary lymph node dissection (ALND) in lymph node staging. However, SLNB was proved to have a not acceptable false negative rate (FNRs) in patients with neoadjuvant chemotherapy (NACT). Thus, it is extremely meaningful to help refine SLNB accuracy.

Analysis is well conducted and many data are present in this review. Nevertheless, I have some small suggestion:

  1. It seemed that the full name of “FNRs” was not given in Simple Summary when it first proposed. (In Simple Summary)

Response: we added the explanation of the acronym FNR in the abstract, as suggested.

  1. Maybe, we’d better use immunohistochemistry staining for further identification of resected LNs. (In 2.2 Pathological evaluation)

Response: We would like to thank the reviewer for this observation. We modified the corresponding materials and methods section to better explain the methods of pathological examination of the lymph nodes:

“SLNs, CLN and all resected LNs were submitted fresh for intraoperative macroscopic and cytologic evaluation. The intraoperative cytology examination of the lymph nodes was performed by dissecting them in two parts along the long axis. After sectioning them, a slide was applied or dragged over the cut surface of both halves and stained with Harris Hematoxylin solution. In case of suspected cytology, lymph node halves were frozen to −22 °C, serially divided in ultrathin sections and stained with Harris hematoxylin solution. In the cases with negative findings, the LNs were completely used up with sectioning at steps of 200 mm with 2 consecutive slides stained with hematoxylin and eosin and pan-cytokeratin (AE1/AE3) immunohistochemistry. The LNs metastasis classification followed the 7th American Joint Committee on Cancer (2010)21: after NACT also isolated tumor cells and micrometastases were considered node positive. The histopathologic results of CLN, SLNs and all resected LNs were given separately in the final pathology report.”

  1. Evidence demonstrated that fluorescence imaging for axillary SLN identification with indocyanine green (ICG) or double labelling are superior to the single technique with blue-dye (In 2.1 Study population) ([1, 2]). If available, using double tracer technique as control group would be better in further study.
  2. Kedrzycki, M.S., et al., Meta-analysis Comparing Fluorescence Imaging with Radioisotope and Blue Dye-Guided Sentinel Node Identification for Breast Cancer Surgery. Ann Surg Oncol, 2021. 28(7): p. 3738-3748.

  1.     Almhanedi, H., et al., Novel double injection technique for sentinel lymph node biopsy in oral cancer. Br J Oral Maxillofac Surg, 2021. 59(10): p. 1296-1301.

Response: We would like to thank the reviewer for this observation and for pointing out these interesting works. We modified the discussion section according to your suggestions and added the corresponding references (29, 30).

“This result is probably related to the relatively high FNR of SLNB in our series (18.75%). Evidence demonstrated that fluorescence imaging for axillary SLN identification with indocyanine green (ICG) or double labelling are superior to the single technique with blue-dye [29, 30] so this may have affect FNR of SLNB in our series.”

  1. In addition, whether the ROLL is benefit for the further treatment decision needed further study.

Response: we thank the reviewer for bringing up this point. We modified the conclusions, as suggested:

“Additional investigations are required to confirm that ROLL-procedure is benefit for the further treatment decision and to identify subsets of patients where ALND can be safely omitted based on this approach.”

Reviewer 2 Report

The presented work investigates a technique for targeted lymph node removal after NACT. This is a highly topical and much discussed subject. Various techniques have already been investigated and published. The ROLL technique for targeted axillary lymph node removal represents an innovative procedure. The manuscript is very well structured and understandable. There are only a few minor comments:

1. In Table 1, "hybrid" is mentioned under phenotype. This designation is not generally understandable. A better classification would be HR+/HER2-, HR-/HER2-, HR+HER2+, HR-/Her2+.

2. Discussion: Here the advantages of the one-step technique are emphasized, which is correct. However, carbon tattooing and radar markers should also be mentioned here as a one-step technique, in addition to iodine seeds and magnetic seeds.

3. Discussion: FNR was certainly important initially to establish targeted lymph node removal. However, whether FNR has an impact on oncologic outcome is currently completely unclear, but of critical interest. This should be addressed in the discussion. The AXSANA study will provide results in this regard.

Author Response

Dear Editor,

Thank you for your email enclosing the reviewers’ comments.

We have carefully reviewed the comments and have revised the manuscript accordingly. Our

responses are given in a point-by-point manner below.

The authors would like to thank the reviewers for their precious time and invaluable comments.

We really hope these modifications can meet with your approval.

REVIEWER 2

The presented work investigates a technique for targeted lymph node removal after NACT. This is a highly topical and much discussed subject. Various techniques have already been investigated and published. The ROLL technique for targeted axillary lymph node removal represents an innovative procedure. The manuscript is very well structured and understandable. There are only a few minor comments:

  1. In Table 1, "hybrid" is mentioned under phenotype. This designation is not generally understandable. A better classification would be HR+/HER2-, HR-/HER2-, HR+HER2+, HR-/Her2+.

Response: we would like to thank the reviewer for this suggestion. We modified Table 1 accordingly.

  1. Discussion: Here the advantages of the one-step technique are emphasized, which is correct. However, carbon tattooing and radar markers should also be mentioned here as a one-step technique, in addition to iodine seeds and magnetic seeds.

Response: we would like to thank the reviewer for this observation. We modified the discussion section to add also the suggested techniques:

“The other drawback of this technique is the double step (clip positioning before NACT and clip localization after NACT) while it would be desirable a one-step procedure. In this regard, iodine-125 seeds (MARI procedure) [20,21] is a possibility or, even better, the use of magnetic seeds [18] or radar markers [19], which also avoid the use of radioactive tracers. Also carbon tattooing was demonstrated to be a simple and less expensive method, with no necessity of extra radiological imaging or nuclear medicine procedures to localize it [16,17,27]. However, ROLL is a feasible and effective method for hospitals that cannot use iodine-125 or magnetic seeds.”

  1. Discussion: FNR was certainly important initially to establish targeted lymph node removal. However, whether FNR has an impact on oncologic outcome is currently completely unclear, but of critical interest. This should be addressed in the discussion. The AXSANA study will provide results in this regard.

 Response: we would like to thank the reviewer for this observation. We modified the discussion section to add this observation:

“However, it need to be highlight that the impact of FNR of SLNB on oncologic outcome is still unclear and it seems to not correspond to an increase in axillary recurrences or a worsening of disease-free survival or overall survival [36]. Ongoing studies, such as AXANA (a large prospective, non-interventional cohort study), aim to demonstrate which is the best method of axillary surgery after NACT in cases starting with a positive axilla, with primary endpoints invasive disease-free survival, axillary recurrence rate and quality of life and arm morbidity [37].”